# Colorectal Neoplasia Detection Rates in Lynch Syndrome

**DOI:** 10.3390/cancers16234021

**Published:** 2024-11-30

**Authors:** Danielle Mirda, Michaela Dungan, Yue Ren, Hongzhe Li, Bryson W. Katona

**Affiliations:** 1Department of Medicine, Hospital of the University of Pennsylvania, Philadelphia, PA 19104, USA; danielle.mirda@pennmedicine.upenn.edu; 2Division of Gastroenterology and Hepatology, Perelman School of Medicine, University of Pennsylvania, Philadelphia, PA 19104, USA; michaela.dungan@pennmedicine.upenn.edu; 3Department of Biostatistics, Epidemiology, and Informatics, Perelman School of Medicine, University of Pennsylvania, Philadelphia, PA 19104, USA; yueren@pennmedicine.upenn.edu (Y.R.); hongzhe@pennmedicine.upenn.edu (H.L.)

**Keywords:** Lynch syndrome, colonic neoplasia, colorectal cancer, colorectal polyps

## Abstract

Individuals with Lynch syndrome (LS) are at increased risk of colorectal cancer and undergo regular lower endoscopic procedures to detect and remove precancerous lesions. However, the frequency of colonic neoplasia detection in LS is not well characterized. In this study, we analyzed over 1200 colonoscopies and sigmoidoscopies from 352 individuals with LS. Detection rates for adenomas and other colorectal lesions were influenced by factors such as age and clinical history, while genotype, sex, and race showed no association. Our findings may provide benchmarks for endoscopists caring for individuals with LS and offer insight into lesion detection patterns. Further research is needed to assess how these rates affect colorectal cancer risk and outcomes, with the goal of optimizing surveillance strategies for this high-risk population.

## 1. Introduction

Lynch syndrome (LS) is the most common hereditary cause of colorectal cancer (CRC) accounting for up to 5% of all CRCs [1]. LS is caused by pathogenic germline variants (PGVs) in DNA mismatch repair (MMR) genes (*MLH1*, *MSH2*, *MSH6*, and *PMS2*) or *EPCAM* (leading to *MSH2* silencing) and leads to earlier development of colonic adenomas, which more often have high-grade dysplasia and villous histology compared to the general population [2]. Additionally, individuals with LS have an accelerated adenoma-carcinoma sequence, as well as the potential to develop CRC directly from MMR-deficient crypts [3,4,5]. Overall, LS confers a lifetime risk of CRC up to 60%; however, this varies by genotype, with *MSH6* and *PMS2* PGV carriers having lower CRC risk compared to those carrying a *MLH1*, *MSH2*, or *EPCAM* PGVs [6].

Surveillance colonoscopy has become the cornerstone of CRC risk management strategies in LS, providing the opportunity for early detection and removal of pre-cancerous lesions, as well as offering the opportunity to identify asymptomatic interval cancers to allow for curative resection. Current NCCN guidelines recommend CRC surveillance in LS every one to two years beginning between ages 20 and 25 for high-risk variants (*MLH1*, *MSH2*, and *EPCAM*) and every one to three years beginning between ages 30 and 35 for *MSH6* and *PMS2* carriers [7]. In LS, the importance of frequent surveillance is evidenced by earlier stages of CRC at diagnosis for those undergoing surveillance [8]. Additionally, CRC identified at early stages is frequently curable, affording a significant decrease in CRC mortality in those who undergo regular surveillance [8,9,10,11].

Given the increased risk of CRC and the proven benefit of surveillance colonoscopy in LS, it is likely that accurate detection and removal of colonic neoplasia in LS is a critical component of a comprehensive cancer risk management strategy for these individuals. However, unlike for the average-risk screening population, there are no well-defined metrics in LS whereby endoscopists can compare their rates of colonic neoplasia detection. In the average-risk population, one frequently utilized metric is the adenoma detection rate (ADR), which is defined as the percentage of average-risk screening colonoscopies where one or more adenomas is detected and serves as one of the most important quality measures for colonoscopy screening [12]. Currently, for the general population, the American Gastroenterological Association (AGA) recommends an ADR of at least 30% in men and 20% in women over age 50 [13], and there have been multiple studies showing that higher ADR reduces subsequent CRC risk and mortality [14,15,16,17]. A landmark study by Corley and colleagues revealed a 3% reduction in subsequent CRC incidence and a 5% reduction in subsequent CRC mortality for every 1% increase in an endoscopist’s ADR [16].

There have been several European studies that have reported colonic neoplasia rates in LS. These studies, which include both prospective data and retrospective registry data, identified an ADR of 11–28%, sessile serrated lesion (SSL) detection rate of 7–8%, and CRC detection rate of 3% in LS [18,19,20,21,22,23,24]. Additional European studies have evaluated the use of chromoendoscopy and reported ADRs ranging from 30–34% [25,26]. Few studies have been conducted specifically assessing colorectal neoplasia detection in a US-based LS population and they have been limited by small sample sizes [27,28]. To aid in helping to define the expected rates of colorectal neoplasia detection in LS, in this study we aimed to determine the overall colorectal neoplasia detection rate (CNDR), the ADR, the proximal serrated detection rate (PSDR), and the CRC detection rate (CRCDR) in individuals with LS and characterize the demographic and clinical factors associated with colorectal neoplasia in this high-risk population.

## 2. Methods

### 2.1. Study Population

This study was approved by the Institutional Review Board of the University of Pennsylvania. In this retrospective study, we identified individuals with LS who were evaluated at Penn Medicine between May 2001 and September 2023. All individuals included had a confirmed PGV in *MLH1*, *MSH2/EPCAM*, *MSH6*, or *PMS2*, or were an obligate carrier of a PGV in one of these genes. Only those individuals with at least one colonoscopy/sigmoidoscopy performed at Penn Medicine within the study period were included for further analysis (Figure 1). Those who did not have a colonoscopy or sigmoidoscopy performed at Penn Medicine were excluded from the analysis (*n* = 190). Prior colorectal surgery was not exclusionary as long as there was remaining colonic tissue that was surveyed with either colonoscopy or sigmoidoscopy. All procedures were performed with high-definition white light endoscopy without the use of dye-based chromoendoscopy.

### 2.2. Study Design

Individual demographic, personal medical history, family history, as well as colonoscopy/sigmoidoscopy and corresponding pathology data were compiled into a secure REDCap database. As our center is a referral center for the coordination of LS care, many of the individuals followed had surveillance endoscopies performed locally and, therefore, data from surveillance endoscopies performed outside of our center were excluded from the study. Additionally, only colonoscopies/sigmoidoscopies performed after LS diagnosis were reviewed. Of the reviewed colonoscopies/sigmoidoscopies, only those that were completed, defined by identification of the ileocecal valve and appendiceal orifice or the ileocolonic anastomosis (if applicable), and having at least adequate prep as defined by the performing endoscopist in the procedure report were included (Figure 1). All of the included sigmoidoscopies were performed in patients with a history of prior colonic resection, such that all remaining colonic mucosa was able to be visualized during the procedure. Information regarding neoplasia size, location, histology, and number was recorded by one single researcher to ensure consistency in data acquisition.

### 2.3. Statistical Analysis

The primary outcome of this study was the endoscopic detection rate of colorectal neoplasia. The detection rates were calculated as the proportion of colonoscopies/sigmoidoscopies with at least one neoplastic lesion divided by the total number of colonoscopies/sigmoidoscopies performed. As there are no well-established neoplasia detection benchmarks in LS, we analyzed multiple different colonic neoplasia detection rates. The detection rates were stratified by lesion type as follows: overall colorectal neoplasia rate (CNDR), which included any CRC, adenoma, or serrated lesion proximal to the sigmoid colon; adenoma detection rate (ADR), which included advanced and non-advanced adenomas; proximal serrated detection rate (PSDR), which included advanced and non-advanced serrated lesions including hyperplastic polyps proximal to the sigmoid colon; and CRC detection rate (CRCDR), which included only biopsy-proven CRCs. Advanced adenomas were defined by size ≥10 mm, tubulovillous or villous histology, or the presence of high-grade dysplasia. Advanced serrated lesions were defined as any serrated lesion ≥10 mm or any traditional serrated adenoma regardless of size.

The secondary outcome was the association of demographic, clinical, or familial risk factors with the presence of colorectal neoplasia. This analysis was performed on a per individual basis with each individual counted only once regardless of the number of colonoscopies/sigmoidoscopies performed. Fisher’s exact test and the Kruskal–Wallis test were used to assess associations between risk factors and the presence or absence of overall colorectal neoplasia, adenomas, and serrated lesions. These analyses were performed using R version 4.2.2. The *p*-value was calculated using a generalized linear mixed-effects model. A *p*-value < 0.05 was considered statistically significant.

## 3. Results

### 3.1. Cohort Demographics and Characteristics

A total of 542 individuals with LS were identified during the study period, with 190 individuals (35.0%) excluded as they did not have at least one colonoscopy/sigmoidoscopy performed at the study site (Figure 1). Thus, a total of 352 individuals who underwent 1323 colonoscopies/sigmoidoscopies performed at the study center were reviewed. Twenty-seven colonoscopies/sigmoidoscopies were excluded as they were incomplete exams due to inadequate prep (98.9% of all procedures had adequate prep) or the inability to examine the entire colonic mucosa for another reason, leaving a total of 1296 procedures (1131 colonoscopies and 165 sigmoidoscopies) included in the analysis (Figure 1). The cohort was primarily white (87.5%), female (64.5%), married (67.9%), privately insured (76.1%), and had never smoked (68.2%) (Table 1). The median age was 48 (interquartile range [IQR], 37–61). The zip code of residence was used as a proxy for income, with 60.2% of individuals residing in a zip code with a median income greater than $100,000.

There was a near even distribution of LS genes, with *MSHS2*/*EPCAM* being the most common genotype (31.8%), followed by *PMS2* (25.6%), *MLH1* (21.6%), and *MSH6* (21.0%) (Table 1). A prior history of CRC was present in 21.9% of the cohort, 26.4% had undergone a prior colonic resection, and 31.5% had a history of a non-colorectal malignancy. Aspirin was used for 2 years or longer in 41.2%. The median endoscopic surveillance interval was 1.2 years (IQR, 1.0–1.7).

### 3.2. Neoplasia Detection Rates

In our cohort of 352 individuals with LS and 1296 surveillance colonoscopies/sigmoidoscopies, colorectal neoplasia was detected on 361 lower endoscopic procedures, leading to an overall colorectal neoplasia detection rate (CNDR) of 27.8% (Figure 2A). At least one advanced adenoma was detected in 5.4% of lower endoscopic procedures, and at least one non-advanced adenoma was detected in 17.9%, for an overall adenoma detection rate (ADR) of 21.4%. Advanced serrated lesions were detected in 0.7% of lower endoscopic procedures, non-advanced serrated lesions in 5.1%, and hyperplastic polyps proximal to the sigmoid colon in 2.4%, for an overall proximal serrated detection rate (PSDR) of 7.7%. There were a total of 18 CRCs detected during surveillance, for a CRC detection rate (CRCDR) of 1.5%.

Next, the colorectal neoplasia rates were stratified based on age at the time of the lower endoscopic procedure (Figure 2B and Appendix A). CNDR increased with increasing age, with colonic neoplasia detected in 13.9% of lower endoscopic procedures performed between ages 30 and 39 compared to 41.8% of those performed at age 60 or older (Figure 2B). Similar trends were noted for adenoma detection, with ADR increasing from 19.3% between ages 30 and 39 to 34.9% at age 60 or older. For CNDR and ADR, individuals with procedures performed under age 30 had a higher rate of neoplasia detection compared to those with procedures performed between ages 30 and 49; however, these results should be interpreted cautiously given the low number of procedures in the under 30 age group. Unlike CNDR and ADR, PSDR was similar across the different age groups (Figure 2B).

### 3.3. Factors Associated with Colorectal Neoplasia

In the generalized linear mixed-effects analysis, older age was associated with both the presence of any colorectal neoplasia and the presence of adenomas (*p* < 0.01) (Table 2 and Table 3). Medicare insurance, history of prior colonic resection, and prior history of non-colorectal malignancy were all statistically significantly associated with the detection of both colorectal neoplasia and adenomas (*p* < 0.05). Additionally, individuals who underwent more surveillance procedures were not surprisingly more likely to have colonic neoplasia and adenomas detected (*p* < 0.01). Higher BMI at first colonoscopy and personal history of prior CRC were associated with adenoma detection (Table 3); however, these were not associated with overall colorectal neoplasia (Table 2). The presence of colorectal neoplasia and adenomas was not significantly associated with genotype, biological sex, race, median income of residence zip code, smoking status, aspirin use (≥2 years), nor family history (*p* > 0.05). There was no significant association between the above risk factors and the presence of serrated lesions (*p* > 0.05) (Appendix A).

### 3.4. Surveillance-Detected Colorectal Cancers

Of the individuals in whom CRC was detected during surveillance, 94.5% were carriers of a PGV in a high-risk gene (*MLH1* or *MSH2/EPCAM*) (Table 4). Of these individuals, 72.2% were over age 50 when CRC was detected during surveillance. CRC was detected during the first surveillance colonoscopy in 27.8% of individuals; however, the majority of individuals with CRC detected during surveillance had colorectal neoplasia on a prior lower endoscopic exam (55.6%). In this cohort, the median time since the last lower endoscopic procedure was completed prior to CRC diagnosis was 1.2 years.

## 4. Discussion

ADR has become a critically important quality metric for endoscopists performing CRC screening in average-risk individuals. However, there is currently no such well-established quality benchmark for LS, which is the most common hereditary CRC risk syndrome and confers a very high lifetime CRC risk. In this retrospective cohort study of 352 individuals with LS who underwent 1296 lower endoscopic surveillance exams, we determined the colorectal neoplasia detection rates, stratified these rates across different age groups, and identified factors associated with detection. This study is one of only a few studies to date that have assessed colorectal neoplasia detection rates in LS, and to our knowledge, it is the largest-to-date conducted in a US LS population [19,20,21,22,23,24,25,26,27].

The overall ADR in our LS cohort under surveillance was 21.4%. This ADR was lower than the recommended ADR in the average-risk population [13]. However, it is important to recognize that individuals with LS typically undergo lower endoscopic surveillance every 1–2 years (every 1.2 years in our cohort – median, IQR 1.0–1.7) and therefore a lower ADR may be expected compared to the average-risk population where colonoscopy is only performed every 10 years. Nonetheless, having a “ballpark” ADR is important for LS and should be helpful for endoscopists who frequently perform lower endoscopic surveillance procedures in individuals with LS.

Importantly, we showed that similar to the average-risk population [29], ADR increased substantially with age in LS. One European study published in 2008 reported a stepwise increase in ADR with increasing age, with a 5% ADR in those aged 16–34 years and a 19% ADR in individuals aged 55–75 years [21]. In our study, individuals aged 60 and older had an adenoma detected more than 4 times more often than those between ages 30 and 39 (34.9% compared to 8.4%). Although we found a similar increase in ADR with increasing age, our rates were significantly higher, likely due to advances in endoscopic equipment and a greater emphasis on polyp detection over the last 15 years. While current ADR metrics for average-risk CRC screening populations are only applicable to those aged 45 and older who qualify for CRC screening, if such an ADR quality metric is implemented in LS, there will likely have to be stratification based on age given that colorectal surveillance starts so much earlier in LS, with drastically different rates of adenoma detection across the age spectrum.

One unexpected finding in our study was that individuals under age 30 had a higher ADR and CNDR compared to those aged 30–49. It is possible that the increased neoplasia rates in individuals under age 30 are attributable to the less frequent surveillance these individuals may undergo, with longer intervals between lower endoscopic procedures, compared to older individuals. Additionally, many young individuals may be diagnosed with LS through cascade testing, and therefore may not have had a colonoscopy prior to their diagnosis. In these cases, their initial surveillance colonoscopy would be their first ever colonoscopy, which could also account for some of the increased neoplasia detection. Additionally, there was the smallest number of colonoscopies/sigmoidoscopies amongst those under age 30 compared to all other age groups, and therefore, given the small sample size, these data should be interpreted with caution.

In addition to ADR, we also examined other colorectal neoplasia rates, including CNDR and PSDR. As detection of all colorectal neoplasia in LS may be important given the elevated CRC risk, we found that the CNDR of our cohort was 27.8%. Similar to ADR, CNDR increased with age, with individuals aged 60 and older having a CNDR approximately three times higher than those between the ages of 30 and 39 years. This trend was not surprising as CNDR is primarily driven by ADR. However, interestingly, PSDR was relatively constant regardless of age at the time of the lower endoscopic surveillance procedure.

We also identified risk factors associated with colorectal neoplasia, with adenomas being associated with multiple factors, including age, BMI, prior colonic resection, as well as both prior CRC and prior cancer other than CRC. However, for serrated lesions, there were no other statistically significant associations. Although identification of these factors associated with colorectal neoplasia is important, at this time in LS we do not have enough evidence to substantially alter colonic surveillance intervals based on these risk factors alone. However, with more data in the future, this could potentially be a possibility.

In our cohort, there were also predictably some interval CRCs that occurred. These interval CRCs were almost exclusively in *MLH1* and *MSH2* carriers, similar to other studies [8,10,30,31,32]. These data are important as they highlight that CRC risk in *MSH6* and *PMS2* carriers under regular active surveillance is likely quite low. Importantly, among the 18 individuals who were found to have an interval CRC during surveillance, only 16.7% had a prior lower endoscopic procedure colonoscopy without any type of colorectal neoplasia. Thus, the consideration of prior neoplasia may be necessary to adequately counsel individuals with LS on their interval CRC risk.

Although having “ballpark” colorectal neoplasia rates is important for endoscopists who regularly perform lower endoscopic procedures in individuals with LS, future research needs to investigate whether colorectal neoplasia rates, such as ADR, are associated with subsequent CRC risk in a similar manner as they are in the average-risk population. Given that some cancers may develop from MMR-deficient crypts in LS, bypassing an intervenable polyp stage, it is possible that high ADR may not impact CRC risk in LS as substantially as it does in average-risk individuals. Future large multicenter studies will be needed to sufficiently answer this question, which will be critical to developing appropriate quality metrics for LS endoscopists.

Our study does have limitations, including that the cohort analyzed was predominantly white, female, and had limited geographic diversity given that this study was performed at a single center. Therefore, it is uncertain if these results would be applicable to other LS cohorts. Additionally, we excluded lower endoscopic procedures that were not performed at our center, which decreased the sample size.

## 5. Conclusions

We determined neoplasia detection rates in an LS cohort and our data showed that CNDR and ADR in LS increased with age whereas PSDR remained constant regardless of age. We believe these findings will provide endoscopists with important “ballpark” detection rates that can be considered when performing endoscopies in individuals with LS, as they do differ from those expected in the average-risk population. However, it will be critical for future studies to determine if colorectal neoplasia detection rates are associated with subsequent CRC outcomes in LS, which will potentially allow these rates to then be adopted as a quality metric for LS endoscopy.

## Figures and Tables

**Figure 1 cancers-16-04021-f001:**
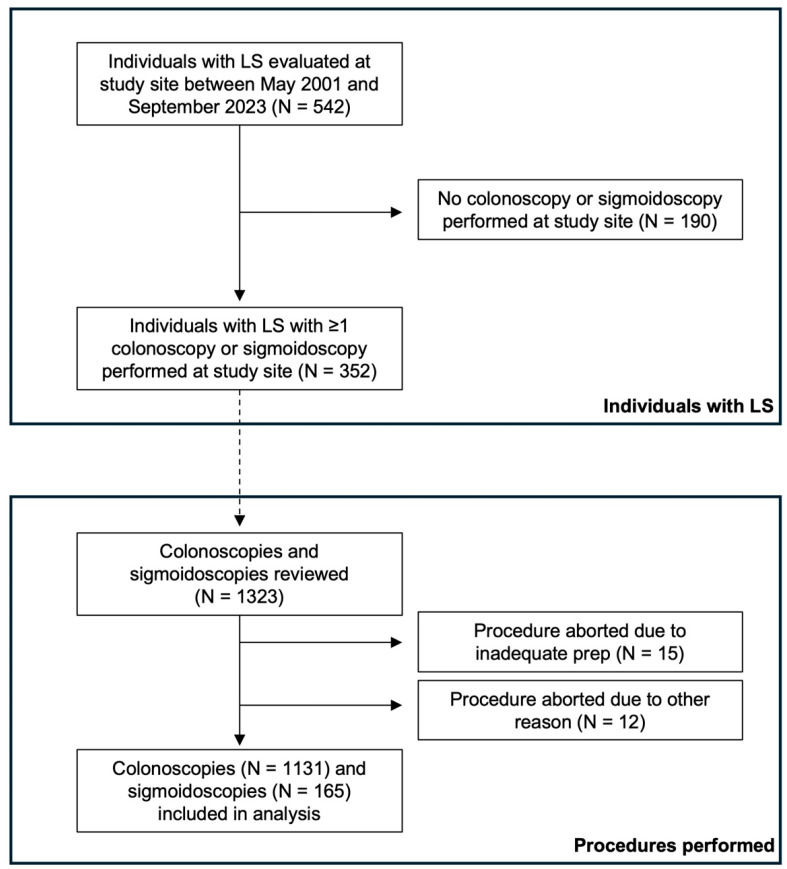
Flowchart of individuals with Lynch syndrome evaluated at the study site and procedures included in the analyses.

**Figure 2 cancers-16-04021-f002:**
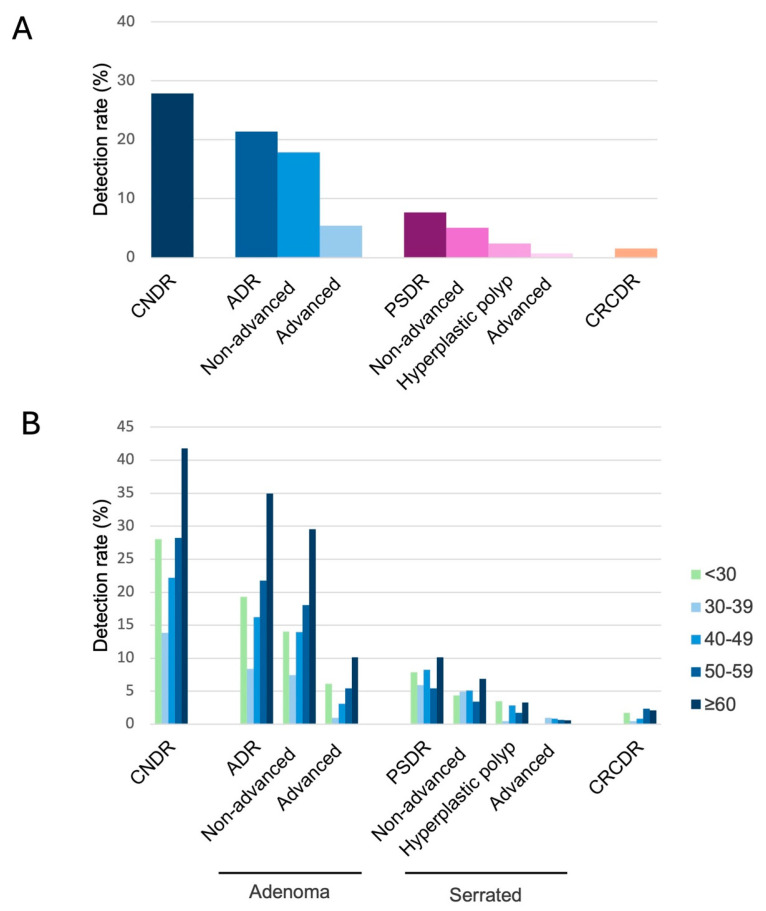
Colorectal neoplasia detection rates, (**A**) overall and (**B**) stratified by age.

**Table 1 cancers-16-04021-t001:** Cohort characteristics.

Variable	*n* = 352
Biological sex (%)	
*Female*	227 (64.5)
	125 (35.5)
Age at first colonoscopy/sigmoidoscopy after LS diagnosis (%)	
*<30*	38 (10.8)
*30–39*	73 (20.7)
*40–49*	76 (21.6)
*50–59*	68 (19.3)
*≥60*	97 (27.6)
Race (%)	
*White*	308 (87.5)
*Black*	12 (3.4)
*Other*	23 (6.5)
*Not reported*	9 (2.6)
Marital Status (%)	
*Single*	77 (21.9)
*Married*	239 (67.9)
*Divorced/Widowed/Other*	36 (10.2)
Insurance Type (%)	
*Private*	268 (76.1)
*Medicare*	69 (19.6)
*Medicaid*	12 (3.4)
*Other*	3 (0.9)
Median income of zip code of residence (%)	
*0–49,999*	17 (4.8)
*50,000–74,999*	43 (12.2)
*75,000–99,999*	80 (22.7)
*100,000–124,999*	121 (34.4)
*125,000–149,999*	50 (14.2)
*≥150,000*	41 (11.6)
Smoking status (%)	
*Never*	240 (68.2)
*Former*	96 (27.3)
*Current*	16 (4.5)
BMI at first colonoscopy/sigmoidoscopy (median [IQR])	26.5 [23.3, 30.8]
ASA use ≥2 years (%)	145 (41.2)
Gene (%)	
*MLH1*	76 (21.6)
*MSH2*/*EPCAM*	112 (31.8)
*MSH6*	74 (21.0)
*PMS2*	90 (25.6)
Surveillance interval in years (median [IQR])	1.2 [1.0, 1.7]
History of prior colon resection (%)	93 (26.4)
Personal history of any prior cancer (%)	188 (53.4)
Personal history of prior colon cancer (%)	77 (21.9)
Personal history of other cancer (%)	146 (41.5)
Family history of any cancer (%)	342 (97.2)
Family history of colon cancer (%)	262 (74.4)
Family history of other cancer (%)	317 (90.1)
Number of colonoscopies/sigmoidoscopies	
*1*	104 (29.5)
*2*	64 (18.2)
*3*	42 (11.9)
*4*	42 (11.9)
*5*	28 (8.0)
*6*	21 (6.0)
*≥7*	51 (14.6)

**Table 2 cancers-16-04021-t002:** Risk factors associated with colorectal neoplasia detection.

Variable	Colorectal Neoplasia Present (*n* = 204)	Colorectal Neoplasia Absent (*n* = 148)	*p*
Biological sex (%)			0.11
*Female*	124 (60.8)	103 (69.6)	
*Male*	80 (39.2)	45 (30.4)	
Median age (%)			<0.01
*<30*	18 (8.8)	20 (13.5)	
*30–39*	22 (10.8)	51 (34.5)	
*40–49*	48 (23.5)	28 (18.9)	
*50–59*	44 (21.6)	24 (16.2)	
*≥60*	72 (35.3)	25 (16.9)	
Race (%)			0.48
*White*	181 (88.7)	127 (85.8)	
*Black*	8 (3.9)	4 (2.7)	
*Other*	10 (4.9)	13 (8.8)	
*Not reported*	5 (2.5)	4 (2.7)	
Marital Status (%)			<0.01
*Single*	34 (16.7)	43 (29.1)	
*Married*	142 (69.6)	97 (65.5)	
*Divorced/Widowed/Other*	28 (13.7)	8 (5.4)	
Insurance Type (%)			0.02
*Private*	148 (72.5)	120 (81.1)	
*Medicare*	49 (24.0)	20 (13.5)	
*Medicaid*	7 (3.4)	5 (3.4)	
*Other*	0 (0.0)	3 (2.0)	
Income (%)			0.52
*0–49,999*	10 (4.9)	7 (4.7)	
*50,000–74,999*	27 (13.2)	16 (10.8)	
*75,000–99,999*	40 (19.6)	40 (27.0)	
*100,000–124,999*	73 (35.8)	48 (32.4)	
*125,000–149,999*	27 (13.2)	23 (15.5)	
*≥150,000*	27 (13.2)	14 (9.5)	
Smoking status (%)			0.13
*Never*	131 (64.2)	109 (73.6)	
*Former*	64 (31.4)	32 (21.6)	
*Current*	9 (4.4)	7 (4.7)	
BMI at first colonoscopy/sigmoidoscopy (median [IQR])	26.5 [24.3, 30.9]	26.3 [22.5, 30.5]	0.28
ASA use ≥2 years (%)	86 (42.2)	59 (39.9)	0.75
Gene (%)			0.58
*MLH1*	45 (22.1)	31 (20.9)	
*MSH2/EPCAM*	70 (34.3)	42 (28.4)	
*MSH6*	41 (20.1)	33 (22.3)	
*PMS2*	48 (23.5)	42 (28.4)	
Surveillance interval in years (median [IQR])	1.1 [1.0, 1.5]	1.2 [1.0, 2.0]	0.12
History of prior colon resection (%)	65 (31.9)	28 (18.9)	0.01
Personal history of any prior cancer (%)	127 (62.3)	61 (41.2)	<0.01
Personal history of prior colon cancer (%)	52 (25.5)	25 (16.9)	0.07
Personal history of other cancer (%)	75 (36.8)	36 (24.3)	0.02
Family history of any cancer (%)	196 (96.1)	146 (98.6)	0.27
Family history of colon cancer (%)	150 (73.5)	112 (75.7)	0.74
Family history of other cancer (%)	46 (22.5)	34 (23.0)	1.00
Number of colonoscopies/sigmoidoscopies			<0.01
*1*	34 (16.7)	70 (47.3)	
*2*	34 (16.7)	30 (20.3)	
*3*	28 (13.7)	14 (9.5)	
*4*	27 (13.2)	15 (10.1)	
*5*	25 (12.3)	3 (2.0)	
*6*	17 (8.3)	4 (2.7)	
*≥7*	37 (19.1)	12 (8.2)	

**Table 3 cancers-16-04021-t003:** Risk factors associated with colorectal adenoma detection.

Variable	Adenoma Present (*n* = 163)	Adenoma Absent (*n* = 189)	*p*
Biological sex (%)			0.05
*Female*	96 (58.9)	131 (69.3)	
*Male*	67 (41.1)	58 (30.7)	
Median age (%)			<0.01
*<30*	12 (7.4)	26 (13.8)	
*30–39*	15 (9.2)	58 (30.7)	
*40–49*	37 (22.7)	39 (20.6)	
*50–59*	31 (19.0)	37 (19.6)	
*≥60*	68 (41.7)	29 (15.3)	
Race (%)			0.09
*White*	146 (89.6)	162 (85.7)	
*Black*	8 (4.9)	4 (2.1)	
*Other*	6 (3.7)	17 (9.0)	
*Not reported*	3 (1.8)	6 (3.2)	
Marital Status (%)			0.01
*Single*	25 (15.3)	52 (27.5)	
*Married*	116 (71.2)	123 (65.1)	
*Divorced/Widowed/Other*	22 (13.5)	14 (7.4)	
Insurance Type (%)			0.01
*Private*	110 (67.5)	158 (83.6)	
*Medicare*	47 (28.8)	22 (11.6)	
*Medicaid*	6 (3.7)	6 (3.2)	
*Other*	0 (0.0)	3 (1.6)	
Income (%)			0.32
*0–49,999*	8 (4.9)	9 (4.8)	
*50,000–74,999*	22 (13.5)	21 (11.1)	
*75,000–99,999*	29 (17.8)	50 (26.5)	
*100,000–124,999*	56 (34.4)	65 (34.4)	
*125,000–149,999*	23 (14.1)	27 (14.3)	
*≥150,000*	24 (14.7)	17 (9.0)	
Smoking status (%)			0.29
*Never*	105 (64.4)	135 (71.4)	
*Former*	51 (31.3)	45 (23.8)	
*Current*	7 (4.3)	9 (4.8)	
BMI at first colonoscopy/sigmoidoscopy (median [IQR])	26.7 [24.5, 31.3]	25.8 [22.7, 29.9]	0.04
ASA use ≥2 years (%)	74 (45.4)	71 (37.6)	0.17
Gene (%)			0.33
*MLH1*	36 (22.1)	40 (21.2)	
*MSH2/EPCAM*	59 (36.2)	53 (28.0)	
*MSH6*	31 (19.0)	43 (22.8)	
*PMS2*	37 (22.7)	53 (28.0)	
Surveillance interval in years (median [IQR])	1.1 [1.0, 1.5]	1.3 [1.0, 2.0]	0.06
History of prior colon resection (%)	56 (34.4)	37 (19.6)	<0.01
Personal history of any prior cancer (%)	107 (65.6)	81 (42.9)	<0.01
Personal history of prior colon cancer (%)	45 (27.6)	32 (16.9)	<0.01
Personal history of other cancer (%)	89 (54.6)	57 (30.2)	0.02
Family history of any cancer (%)	156 (95.7)	186 (98.4)	0.23
Family history of colon cancer (%)	123 (75.5)	139 (73.5)	0.77
Family history of other cancer (%)	150 (92.0)	167 (88.4)	0.33
Number of colonoscopies/sigmoidoscopies			<0.01
*1*	23 (14.1)	81 (42.9)	
*2*	26 (16.0)	38 (20.1)	
*3*	20 (12.3)	22 (11.6)	
*4*	24 (14.7)	18 (9.5)	
*5*	20 (12.3)	8 (4.2)	
*6*	15 (9.2)	6 (3.2)	
*≥7*	22 (13.5)	14 (7.4)	

**Table 4 cancers-16-04021-t004:** Characteristics of individuals with surveillance-detected CRC.

Variable	*n* = 18
Gene (%)	
*MLH1*	7 (38.9)
*MSH2*/*EPCAM*	10 (55.6)
*MSH6*	0 (0.0)
*PMS2*	1 (5.6)
Age at time of colonoscopy/sigmoidoscopy (%)	
*<30*	1 (5.6)
*30–39*	1 (5.6)
*40–49*	3 (16.7)
*50–59*	7 (38.9)
*≥60*	6 (33.3)
Prior colorectal neoplasia (%)	
*No*	3 (16.7)
*Yes*	10 (55.6)
*N/A* *	5 (27.8)
Interval from prior lower endoscopic procedure in years (median [IQR])	1.2 [1.0, 1.4]

* CRC detected on first colonoscopy.

## Data Availability

The data presented in this study are not publicly available due to privacy and ethical restrictions. De-identified data may be made available upon request from the corresponding author.

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
