# Peer review of "Colorectal Neoplasia Detection Rates in Lynch Syndrome"

_cancers, 2024, doi:10.3390/cancers16234021_

Round 1

Reviewer 1 Report

Comments and Suggestions for Authors

Thank You for allowing me to review this interesting cohort study evaluating the detection of pre-cancerous and cancerous lesions using endoscopy in lynch syndrome. this is a topical subject. the manuscript is well written overall, as are the objectives. however, i do have some comments and questions. 

almost a third of the population (190 patients) were excluded because they did not undergo endoscopy. does this mean that they are not being monitored? 

The authors included both a total colonoscopy and a short colonoscopy for the detection of mucosal lesions. In my opinion, this is a major bias because it is not at all the same examination; unless the short colonoscopy was performed in patients who had undergone a subtotal or total colectomy. Furthermore, patients were excluded if the endoscopy was not complete. how can a short colonoscopy be complete? 

in the inclusion criteria, the authors group together cancers and adenomas in the rate of colorectal neoplasia. then the authors include in the rate of adenomas those without and with dysplasia. i confess i don't understand the distinction. 

Finally, was chromoendoscopy performed in the case of neoplastic lesions in order to increase the rate of detection of adenomas? 

to sum up, the joint inclusion of total and short colonoscopies lowers the message of the real detection rate. i would encourage the authors to separate the results according to the endoscopic examinations carried out.

Author Response

We sincerely thank the expert reviewers for their thoughtful critiques and insightful recommendations regarding our manuscript. We hope that the revised version addresses their concerns comprehensively and is deemed suitable for publication in Cancers. Below is our detailed point-by-point response to the reviewers:

Reviewer #1

Thank You for allowing me to review this interesting cohort study evaluating the detection of pre-cancerous and cancerous lesions using endoscopy in lynch syndrome. this is a topical subject. the manuscript is well written overall, as are the objectives. however, i do have some comments and questions. 

Thank you very much for your thoughtful review of our manuscript. I hope we can provide clarification here below to address your comments and questions.

Almost a third of the population (190 patients) were excluded because they did not undergo endoscopy. does this mean that they are not being monitored? 

You have brought up an excellent question -- we excluded these 190 patients because they did not undergo a colonoscopy/sigmoidoscopy at our center (Penn Medicine). We care for many patients at Penn that have their procedures performed at centers other than our center, which are often closer to their homes. Since we do not always have access to these endoscopy and pathology reports, we excluded these patients from our analysis. We have added a sentence in the Methods section under ‘Study Population’ to clarify this.

The authors included both a total colonoscopy and a short colonoscopy for the detection of mucosal lesions. In my opinion, this is a major bias because it is not at all the same examination; unless the short colonoscopy was performed in patients who had undergone a subtotal or total colectomy. Furthermore, patients were excluded if the endoscopy was not complete. how can a short colonoscopy be complete? 

Thank you for raising this concern and we would certainly like to make sure that this point is clarified.  In our cohort, only patients with a history of a prior colonic resection underwent sigmoidoscopies. These sigmoidoscopies were complete in that the endoscopist reached the ileocolonic anastomosis and thus all of the remaining colonic mucosa was visualized. Sigmoidoscopies that were not complete were not included in the analysis.  We have added a sentence under ‘Study Design’ to address this.

In the inclusion criteria, the authors group together cancers and adenomas in the rate of colorectal neoplasia. then the authors include in the rate of adenomas those without and with dysplasia. i confess i don't understand the distinction. 

Thank you for bringing up this point. Given the lack of established colonic neoplasia detection benchmarks in Lynch syndrome, we decided to evaluate a variety of different colonic neoplasia rates amongst our cohort. Our overall colorectal neoplasia rate (CNDR) included all neoplastic lesions of the colon, but we also examined the adenoma detection rate (ADR), and the proximal serrated detection rate (PSDR) which included all serrated lesions including a subset of hyperplastic polyps. Within the categories, we stratified by advanced and non-advanced, based on lesion size and pathology. We added two sentences under the section titled ‘Statistical analysis’ in an effort to clarify this point.

Finally, was chromoendoscopy performed in the case of neoplastic lesions in order to increase the rate of detection of adenomas? 

This is an excellent question and is certainly one we should clarify for readers. Our center does not use dye-based chromoendoscopy for Lynch syndrome surveillance procedures. However, all colonoscopies and sigmoidoscopies were performed with high-definition white light endoscopy. We included this clarification under the section titled ‘Study Population.’

to sum up, the joint inclusion of total and short colonoscopies lowers the message of the real detection rate. i would encourage the authors to separate the results according to the endoscopic examinations carried out.

Thank you for bringing up this important concern. Sigmoidoscopies were performed on individuals with a prior colonic resection, and these individuals, even though they were lacking their entire colon, still had significantly higher rates of colorectal neoplasia detection (Table 2) and higher rates of adenoma detection (Table 3) compared to those with intact colons. Given the high detection rates in this group as well as the fact that sigmoidoscopies included in the analysis were only those allowing for examination of the entire colonic mucosa, we feel strongly that neoplasia detection rates should be calculated using both the sigmoidoscopy and colonoscopy data together.  We hope that helps clarify why we chose this method of analysis.

Reviewer 2 Report

Comments and Suggestions for Authors

1) Table 2: There is no difference in ADR(+)/(-) in the surveillance interval, but there is a difference in ADR(+)/(-) in the number of colonoscopy/sigmoidoscopy. Therefore, shouldn't authors analyze the correlation between follow-up period and ADR?

2) Tables 2, 3: Shouldn't authors perform an analysis of only the total colonoscopy group, excluding the sigmoid colonoscopy group?

3) Tables 2, 3: Shouldn't authors perform a multivariate analysis?

4) Table 2: Why is there no significant difference only in Personal history of prior colon cancer, even though the ADR (+):(-) ratio is almost 2:1 in History of prior colon resection, Personal history of any prior cancer, Personal history of prior colon cancer, and Personal history of other cancer?

Author Response

We sincerely thank the expert reviewers for their thoughtful critiques and insightful recommendations regarding our manuscript. We hope that the revised version addresses their concerns comprehensively and is deemed suitable for publication in Cancers. Below is our detailed point-by-point response to the reviewers:

Reviewer #2

1) Table 2: There is no difference in ADR(+)/(-) in the surveillance interval, but there is a difference in ADR(+)/(-) in the number of colonoscopy/sigmoidoscopy. Therefore, shouldn't authors analyze the correlation between follow-up period and ADR?

Thank you for your comment. We appreciate the opportunity to clarify the distinction between surveillance interval and the number of colonoscopies/sigmoidoscopies. The surveillance interval refers to the time between consecutive surveillance colonoscopies or sigmoidoscopies, which in our study showed no significant difference in those with and without adenomas (1.1 vs. 1.2 years, respectively).  On the other hand, the number of colonoscopies/sigmoidoscopies referred to the total number of surveillance procedures the individual had.  There was a significant difference noted for the total number of surveillance procedures, with those with adenomas having more surveillance procedures on average that those without adenomas.  Even with a similar surveillance interval, this increase in adenoma detection with more procedures performed is expected in our opinion, as there are more opportunities to assess for adenomas over a longer period of time.  Given this, the number of procedures indirectly captures the effect of a longer follow-up period, and therefore we felt it important to keep both measures separate in the analysis. We have included a brief sentence in the manuscript section titled ‘Factors associated with colorectal neoplasia’ to address this.

2) Tables 2, 3: Shouldn't authors perform an analysis of only the total colonoscopy group, excluding the sigmoid colonoscopy group?

Thank you for raising this concern and we apologize for not being clear about our intentions in our original manuscript. Only patients with a history of a prior colon resection underwent sigmoidoscopies. These sigmoidoscopies were complete in that the endoscopist reached the ileocolonic anastomosis and all remaining colonic mucosa was visualized during the examination. We have added a sentence under ‘Study Design’ to address this.

Additionally, individuals with a prior colonic resection (including those that underwent sigmoidoscopy) had high rates of colorectal neoplasia detection (Table 2) and high rates of adenoma detection as well (Table 3). Given the high detection rates in this group, we felt that inclusion of sigmoidoscopy would not artificially decrease the neoplasia detection rates in our cohort, and therefore, we did not believe it was necessary to separate colonoscopy and sigmoidoscopy results in our analysis.

3) Tables 2, 3: Shouldn't authors perform a multivariate analysis?

Thank you for your question regarding the use of multivariate analysis. We certainly understand the value of multivariate approaches for understanding the relationships between multiple variables and outcomes. However, in this study, we opted to use Generalized Linear Mixed-Effects Models rather than traditional multivariate analysis for the following reasons:

  1. Repeated Measures per Patient: Our dataset contains multiple colonoscopies/sigmoidoscopies from individual patients. Using a multivariate approach without accounting for the repeated measurements could have led to the risk of inflating Type I errors due to the assumption of independence between observations. The mixed-effects model we used appropriately accounts for the repeated measures by incorporating a random intercept for each patient, allowing for patient-specific variability in detection rates.
  2. Patient-Level Analysis: Given that multiple colonoscopies/sigmoidoscopies were performed on some patients, we chose to treat each patient as a unit of analysis rather than each individual procedure. By averaging detection rates across procedures per patient, we ensured that the analysis focused on the patient-level associations with factors such as age, genotype, and clinical history. We believe this approach helps avoid potential skewing from repeated procedures.
  3. Fixed and Random Effects: The fixed effects in the mixed-effects model provide estimates of the overall relationships between predictor variables (e.g., age, prior colonic resection) and detection rates, while the random effects account for within-patient correlation. We believe this method provides a robust way to understand the impact of these factors, while adjusting for individual variability.

In conclusion, we appreciate the suggestion of multivariate analysis, but we believe that our chosen statistical approach, Generalized Linear Mixed-Effects Models, is best for this dataset given the repeated measures and patient-level variability.

4) Table 2: Why is there no significant difference only in Personal history of prior colon cancer, even though the ADR (+):(-) ratio is almost 2:1 in History of prior colon resection, Personal history of any prior cancer, Personal history of prior colon cancer, and Personal history of other cancer?

Thank you for your insightful comment. While the ADR (+):(-) ratio for personal history of prior colon cancer appears close to that of other risk factors, it is important to consider both the sample size and the ratio of percentages when comparing these factors. In the case of prior colon cancer, there were only 77 patients, with a ratio of 1.5, compared to the 93 patients with a history of prior colon resection, which had a slightly higher ratio of 1.7 (as an example). This difference in sample size and the ratio of percentages help to explain the lack of statistical significance observed in the analysis for prior colon cancer.

Round 2

Reviewer 1 Report

Comments and Suggestions for Authors

the authors responded point by point to questions and comments designed to improve the quality of the manuscript 

Reviewer 2 Report

Comments and Suggestions for Authors

Questions are well answered.